# Electrochemical C−C bond cleavage of cyclopropanes towards the synthesis of 1,3-difunctionalized molecules

Pan Peng[1], Xingxiu Yan[1], Ke Zhang[1], Zhao Liu[1], Li Zeng[1], Yixuan Chen[1], Heng Zhang[1 ✉] & Aiwen Lei [1,2 ✉]

Electrochemistry has a lot of inherent advantages in organic synthesis and many redox reactions have been achieved under electrochemical condition. However, the electrochemical C−C bond cleavage and functionalization reactions are less studied. Here we develop electrochemical C−C bond cleavage and 1,3-difuntionalization of arylcyclopropanes under catalyst-free and external-oxidant-free conditions. 1,3-difluorination, 1,3-oxyfluorination and 1,3-dioxygenation of arylcyclopropanes are achieved with a high chemo- and regioselectivity by the strategic choice of nucleophiles. This protocol has good functional groups tolerance and can be scaled up. Mechanistic studies demonstrate that arylcyclopropane radical cation obtained from the anode oxidation and the subsequently generated benzyl carbonium are the key intermediates in this transformation. This development provides a scenario for constructing 1,3-difunctionalized molecules.

[1] The Institute for Advanced Studies (IAS), College of Chemistry and Molecular Sciences, Wuhan University, Wuhan, People's Republic of China. [2] King Abdulaziz University, Jeddah, Saudi Arabia. ✉email: hengzhang@whu.edu.cn; aiwenlei@whu.edu.cn

C–C bonds are the basic skeleton of organic compounds and the direct functionalization via C–C bond deconstruction is quite meaningful for synthesis of complex molecules[1,2]. Cyclopropanes are important building blocks. The ring-opening of cyclopropanes driven by the release of ring strain has been widely applied in total synthesis[3,4]. Donor–acceptor cyclopropanes (DACs), which are activated by vicinal electron-donating and electron-withdrawing groups are predisposed to ring-opening under Lewis acid catalysis due to the inherent electronic bias[5–7], whereas non-activated cyclopropanes, which are more regular in nature, are reluctant to ring-opening due to insufficient electronic bias. There are two methods for ring-opening of non-activated cycolpropanes. One relies on oxidative addition by transition metals. However, these reaction are limited to ring-opening rearrangement or cycloaddition reactions and requires specific directing groups for regioselective ring-opening functionalization[8]. The other one relies on electrophilic activation with Lewis acidic species. However, most of the transformations are limited to electrophilic addition reactions[9–17]. Ring-opening functionalization of arylcyclopropanes initiated through the single electron oxidation followed by the yield of corresponding radical cations was discovered in the 1970s[18,19]. In recent years, this strategy was further applied in 1,3-aminofunctionalization, 1,3-oxoamination, and 1,3-oxochlorination of arylcycropropanes in the presence of oxidants or light. However, in most cases, its large scalability application was not studied and arylcycropropanes with electron-withdrawing groups could not be well compatible due to their high oxidative potential (Fig. 1a)[20–24].

Organic electrochemistry is reviving due to their effortlessness of scalability, avoidance of stoichiometric oxidants or reducing agents, and flexible reaction tunability[25]. Various redox reactions have been achieved by the comsumption of traceless electrons under constant potential or current conditions[26–38]. As a main part of preparative electrosynthesis, anode processes such as C–H functionalization, oxidative coupling, decarboxylation, and olefin functionalization has been developed[39–48]. However, electrochemical oxidative C–C bond cleavage/functionalization are rarely developed due to the inertness and weak electronic bias of C–C bonds, which are always encumbered by other bonds[8,48,49]. Pioneering work was disclosed by Shono and coworkers who reported anodic oxidation of arylcyclopanes in methanol[50]. But only six examples were presented in this report (Fig. 1b). Our design for electrochemical C–C bond cleavage/functionalization based on the following mechanistic proposal (Fig. 1c): Firstly, arylcyclopropane is oxidized to a radical cation by anode, which results in the weakening of the $C_\alpha$–$C_\beta$ bond, as the BDE of $C_\alpha$–$C_\beta$ bond decreases more than 30 kcal/mol from the neutral cyclopropane to the corresponding radical cation[51]. Then the radical cation undergoes three-electron $S_N2$ reaction to generate a benzyl radical[52]. Different from reported thermochemical and photochemical strategy, the benzyl radical can further lose one electron at anode and converted to a benzyl carbonium under electrochemical conditions[43]. The following nucleophilic attack to the benzyl carbonium can finally yield the 1,3-difunctionalization product. Fluorinated products could be prepared by employing $Et_3N\cdot3HF$ as nucleophilic fluorine source[53–58]. In this work, we develop the electrochemical 1,3-difluorination, 1,3-oxyfluorination, and 1,3-dioxygenation of arylcyclopropanes with a high chemoselectivity and regioselectivity by the strategic choice of nucleophiles. Moreover, a wide variety of arylcyclopropanes with electron-donating and electron-withdrawing groups could be converted to the 1,3-difunctionalized molecules by following this protocol.

## Results

**Investigation of reaction conditions.** We began our investigation by exploring the selective synthesis of 1,3-difluorination, 1,3-oxyfluorination, and 1,3-dioxygenation products from phenylcyclopropane. After extensive screening of various conditions (for more details, see Supplementary Table 1–5), with the use of $Et_3N\cdot3HF$ as a fluorine source, 1,3-difluorination product **2** was obtained in 77% yield by conducting the electrolysis under constant current of 16 mA in an undivided cell equipped with platinum plate as both anode and cathode (Fig. 2, Entry 1). 1,3-oxyfluorination product **3**

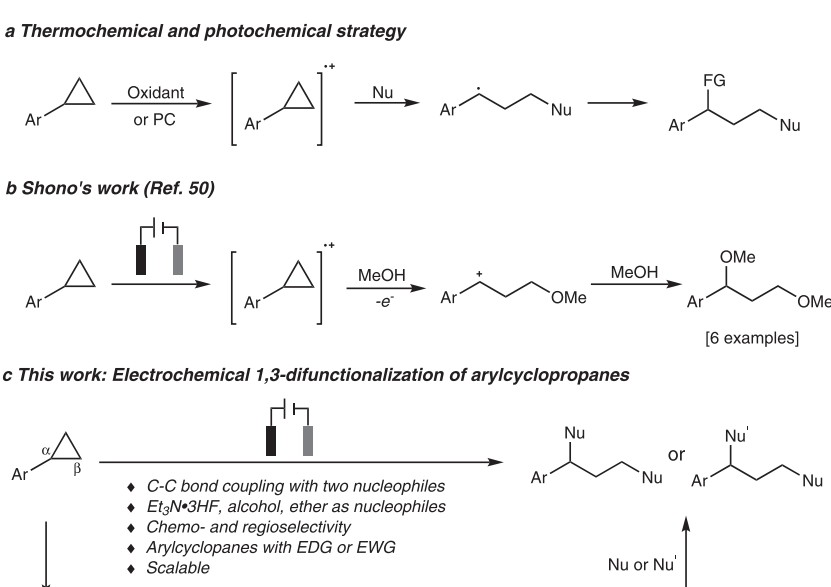

**Fig. 1 1,3-difunctionalization of cyclopropanes based on radical cation mechanism. a** Thermochemical and photochemical strategy. **b** Shono's work. **c** This work: Electrochemical 1,3-difunctionalization of arylcyclopropanes (PC photoredox catalyst).

**Fig. 2 Optimization of the reaction conditions[a].** [a]Reaction conditions: electrode surface ($1.5 \times 1.5$ cm$^2$, $J = 7.1$ mA/cm$^2$), 16 mA, undivided cell, **1** (0.5 mmol), PhCF$_3$ (4.8 mL), 2 h, 2.4 F mol$^{-1}$ (based on **1**). [b]GC Yield using biphenyl as internal standard. [c]**1** (0.25 mmol), 1 h. [d]Without PhCF$_3$, Bu$_4$NBF$_4$ (0.25 mmol) was added as supporting electrolyte (CC Carbon cloth).

| Entry | Temp. (°C) | Et$_3$N·3HF (mL) | CH$_3$OH (mL) | Electrode | Yield (%)[b] | | | |
|---|---|---|---|---|---|---|---|---|
| | | | | | **2** | **3** | **4** | **5** |
| 1 | 25 | 1.2 | 0 | Pt(+)\|Pt(-) | 77 | 0 | 0 | 0 |
| 2 | 25 | 1.2 | 0.5 | Pt(+)\|Pt(-) | 7 | 47 | 4 | 19 |
| 3 | 0 | 1.2 | 0.5 | Pt(+)\|Pt(-) | 7 | 50 | 4 | 20 |
| 4 | 0 | 1.2 | 0.5 | CC(+)\|Pt(-) | 3 | 63 | 2 | 25 |
| 5[c] | 0 | 1.2 | 0.5 | CC(+)\|Pt(-) | 2 | 64 | 2 | 27 |
| 6[c] | 0 | 0.8 | 0.2 | CC(+)\|Pt(-) | 5 | 76 | 2 | 16 |
| 7[c,d] | 25 | 0 | 6 | CC(+)\|Pt(-) | 0 | 0 | 0 | 95 |

was obtained in 47% yield with the concomitant formation of other three 1,3-difuntionalization products under the existence of both Et$_3$N·3HF and MeOH (Fig. 2, Entry 2). The yield of 1,3-oxyfluorination product **3** increased slightly when the reaction temperature decreased to 0 °C (Fig. 2, Entry 3). An obviously improved yield was observed by using carbon cloth as anode materials (Fig. 2, Entry 4). The influence of the concentration of phencyclopropane to the reaction could be neglected (Fig. 2, Entry 5). 1,3-oxyfluorination product **3** was finally observed in 76% yield by adjusting the ratio of Et$_3$N·3HF/MeOH (Fig. 2, Entry 6). The good selectivity of the 1,3-oxyfluorination product **3** was speculated to be controlled by kinetics. At the first step, the reaction rate between MeOH and arylcyclopropane radical cation is larger than the reaction rate between Et$_3$N·3HF and arylcyclopropane radical cation, which possibly due to the lower nucleophilicity of Et$_3$N·3HF than MeOH[57]. Therefore, arylcropropane radical cation mainly reacts with MeOH instead of Et$_3$N·3HF. According to the rate constant equation proposed by Mayr et al, the reaction rate of benzyl carbonium with MeOH or Et$_3$N·3HF are both very fast and determined by diffusion rate[59,60], so the fluorination is the major process in the second step because of the excess amount of fluorine source compared with methanol in the reaction system. Further condition screening demonstrated that 1,3-dioxygenation products could be obtained in 95% yield under the electrolysis in MeOH with Bu$_4$NBF$_4$ as supporting electrolyte (Fig. 2, Entry 7).

**Substrate scope**. To investigate the substrate scope of the 1,3-difluorination reaction, we tested a wide range of arylcyclopropanes under the optimized condition (Fig. 3). Phenylcyclopropane gave 75% NMR yield (**2**). Methanesulfonyl group no matter at *para* or *meta* position was well amenable to this protocol in good yields (**6** and **7**). Halogen groups such as F, Cl, Br, and I were also compatible with this transformation (**8–12**), which provided an opportunity to further transformation through coupling reactions. *Ortho* substituted arylcyclopropane was also suitable reaction partner, furnishing the desired 1,3-difluorination product with 58% yield (**13**). Arylcyclopropanes with electron-withdrawing groups (such as ester, amide, acetyl, cyano, nitro, and aldehyde groups) were also amenable, furnishing the desired

products in 42–56% yields by prolonging the reaction time (**14–20**), although these substrates have higher oxidative potential. However, Arylcyclopropanes with electron-donating groups (such as –OMe, SMe, OPh, and NHBoc) were not amenable in this transformation (for more details, see Supplementary Fig. 3). Trifluoromethyl and oxytrifluoromethyl substituted arylcyclopropanes gave the corresponding products in 49% and 53% yields, respectively (**21–22**). TMS group was retained in this transformation with 52% yield (**23**). Alkyl group such as tert-butyl group substituted arylcyclopropanes underwent the reaction in lower yield of 40% (**24**). 3-Methyl-4-ester substituted arylcyclopropane provided the desired product with 54% yield (**25**). Moreover, heterocyclic arylcyclopropane such as thienyl cyclopropane was also suitable for this reaction (**26**). Then, the feasibility of 1,1-disubstituted cyclopropanes and 1,2-disubstituted cyclopropanes were also tested. 1,1-disubstituted cyclopropanes were good reaction partners to this reaction, giving the 1,3-difluorination products with tertiary-carbon-fluorine bonds formation in good yields (**27–29**). *Trans*-1,2-diphenylcyclopropane provided 1,3-difluorination products with 61% yield and 1.3:1 diastereoselectivity (**30**) since 1,2-diphenylcyclopropane radical cation was once determined to be open geometries[51]. Trisubstituted cyclopropane was also tolerated in this transformation (**31**). Moreover, arylcyclopropanes containing complex nature product scaffolds were also compatible in this transformation. L-menthol scaffold substrate provided the desired product in 86% yield (**32**). 5$\alpha$-cholestan-3$\beta$-ol scaffold substrate provided the coresponding product in 65% yield with 10% starting material recoverd (**33**). Protected sugar group was also compatible with 82% yield was obtained (**34**). Androsterone scaffold substrate gave the desired product in 33% yield with 36% starting material recoverd (**35**). The compatibility of these complex substrates further confirms the practicality of this transformation.

The scalability of this method was demonstrated by using carbon cloth as anode and nickel foam as cathode instead of expensive platinum electrode with equivalent amount of Et$_3$N·3HF (Fig. 3). The yields increased for most of the tested substrates when operated at gram scale. The yields increased from 75% to 90% for *para*-methanesulfonyl substituted phenylcyclopropane (**6**). The yields increased to 65% and 63% for ester (**14**)

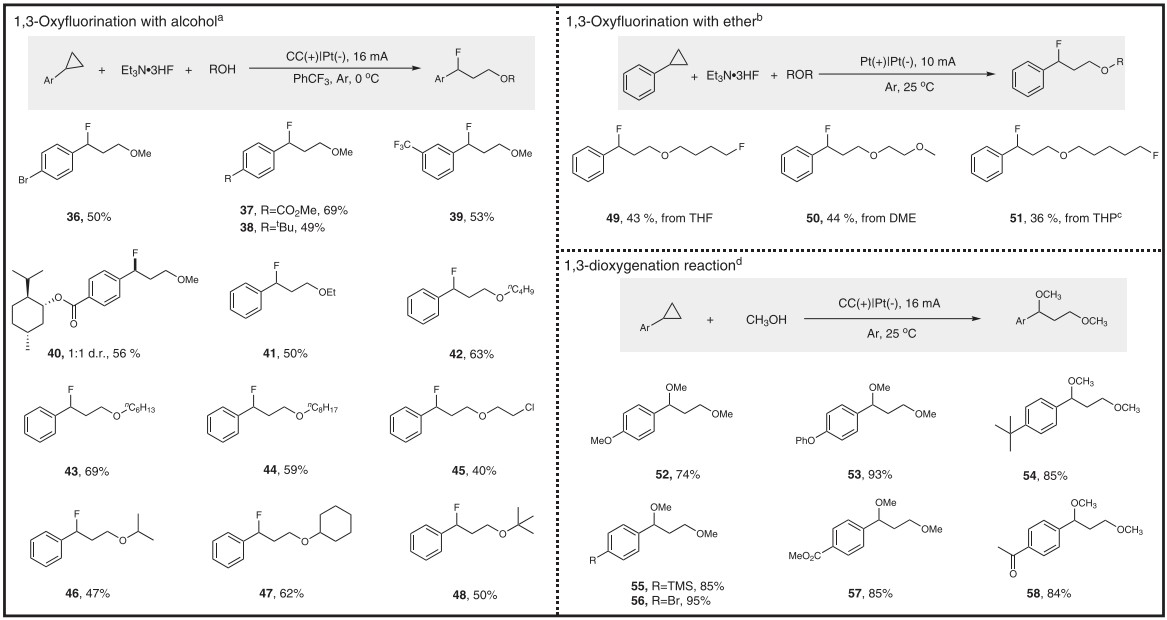

**Fig. 3 Substrate scope of 1,3-difluorination reaction.** Reaction condition: Cyclopropanes (0.25 or 0.5 mmol), Et$_3$N·3HF (1.2 mL), PhCF$_3$ (4.8 mL), 16 mA, 40 min–20 h, isolated yields, electrode surface (1.5 × 1.5 cm$^2$, $J$ = 7.1 mA/cm$^2$), undivided cell. [a]NMR yield using 1-fluoronaphthalene as internal standard. [b]0.5 mL DCE was added. [c]0.1 mmol scale, 2–6 h. s.m. starting material. [d]Large-scale synthesis condition: carbon cloth anode, nickel foam cathode, cyclopropanes (6 or 8 mmol), Et$_3$N·3HF (2.4 mL), PhCF$_3$ (9.6 mL), 25–75 h, isolated yields. For more details, see Supplementary Experimental section.

**Fig. 4 Substrate scope of 1,3-oxyfluorination and 1,3-dioxygenation reaction.** [a]Cyclopropanes (0.25 mmol), Et$_3$N·3HF (0.8 mL), ROH (0.2–0.75 mL), PhCF$_3$ (4.8 mL), 16 mA, 1–2 h, isolated yields. [b]Cyclopropanes (0.5 mmol), Et$_3$N·3HF (1.2 mL), ether (4.8 mL), 4 h, isolated yields. [c] One milliliter of DCE was added. [d]Cyclopropanes (0.25 mmol), Bu$_4$NBF$_4$ (0.25 mmol), MeOH (6 mL), 16 mA, 50 min–1h, isolated yields. For more details, see Supplementary Experimental section.

and oxytrifluoromethyl (**22**) substituted arylcyclopropanes. The yields would decrease slightly from 67% to 63% only when it came to 4-bromocyclopropylbenzene (**10**).

The scope of the 1,3-oxyfluorination reaction was also explored, as shown in Fig. 4. *Para*-bromine substituted phenylcyclopropane was compatible to give the target product in moderate yield (**36**). Both electro-donating group and electro-withdrawing groups could be tolerated in this transformation (**37–40**). *Meta*-trifluoromethyl substituted arylcyclopropane generated 1,3-oxyfluorination product **39** in 53% yield. This 1,3-oxyfluorination product could be potentially converted to fluorinated Cinacalcet following the reported methods[61,62]. Subsequently, the scope of the alcohols were examined as well. Alcohols such as EtOH, *n*-BuOH and long-chain primary alcohols (*n*-C₆H₁₃OH and *n*-C₈H₁₇OH) were transformed into desired products with good efficiency (**41–44**). 2-Chloro-1-ethanol afforded the desired product in 40% yield (**45**). Secondary alcohols (*i*-PrOH and cyclohexanol) were suitable substrates (**46–47**). Sterically hindered tertiary alcohol was also tolerant (**48**). In addition to alcohols, ethers could also participate in this reaction (Fig. 4). 1,3-Oxyfluorination could proceed smoothly when using ether as both oxygen nucleophiles and solvent (**49–51**). Cyclic ether such as tetrahydrofuran gave interesting product **49** in 43% yields accompanying with 14%

1,3-difluorination product. 1,2-Dimethoxyethane gave 1,3-oxy-fluorination product **50** in 44% yield. The reaction system was separated into two phases and not conductive when using tetrahydropyran as both reactant and solvent. To solve the problem of solubility and conductivity, DCE was added to enhance the solubility and 1,3-oxyfluorination product **51** was obtained in 36% yield. Notably, C–O etheric bond was broken in these transformations. We proposed that oxonium is the key intermediate for cleavage of C–O etheric bond (for more details, see Supplementary Fig. 5)[63,64]. Finally, the scope of the 1,3-dioxygenation reaction was investigated (Fig. 4). Arylcyclopropanes with electron-donating groups (–OMe, –OPh) gave 74% and 93% yields, respectively (**52–53**). *Para* tert-butyl substituted arylcyclopropane underwent the reaction in 85% yield (**54**). Both TMS and halogen groups could be tolerated (**55–56**). The electronically deficient arylcyclopropane could also give 1,3-dioxygenation products with high yields (**57–58**).

## Discussion

To gain more insights into the aforementioned transformation, several mechanistic studies were conducted. Redox potentials of the cyclopropanes were tested by cyclic voltammetry experiments (Fig. 5a). The alkyl substituted cyclopropane **1c** has a high

### *a Cyclic voltamemetry studies*

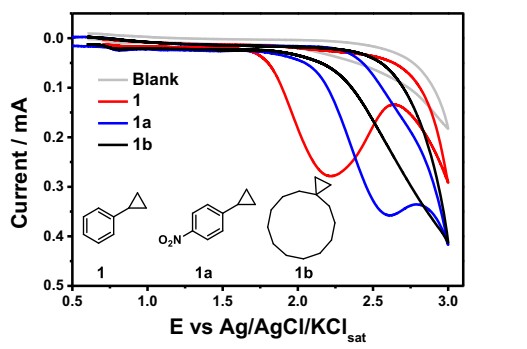

### *b Electrostatic potential surface and charge of 1·⁺*

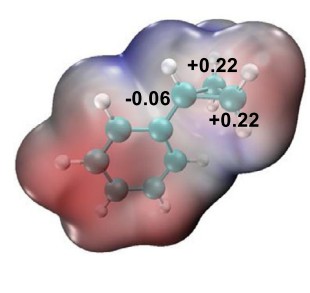

### *c Trap of benzyl radical intermediate*

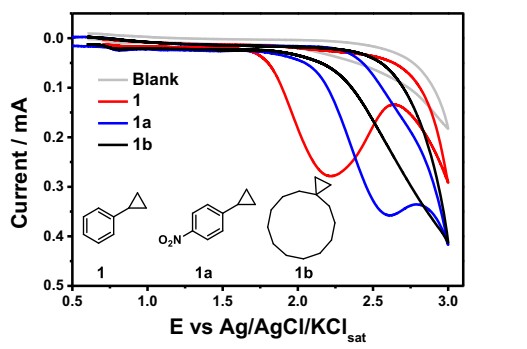

### *d Trap of benzyl carbonium intermediate*

**59**, 43%

**Fig. 5 Mechanistic studies. a** Cyclic voltamemetry studies. **b** Electrostatic potential surface and charge of phenylcyclopropane radical cation (**1·⁺**). **c** Trap of benzyl radical intermediate. **d** Trap of benzyl carbonium intermediate.

oxidation potential and no oxidation peak can be observed under 3 V. In contrast, arylcyclopropanes even with strong electron with-drawing group ($-NO_2$) have relatively lower oxidative potentials ($E_{p/2} = 1.96$ V vs. Ag/AgCl/KCl$_{sat}$ for **1** and $E_{p/2} = 2.32$ V for **1a** vs. Ag/AgCl/KCl$_{sat}$). These results illustrate that aryl group is crucial for the oxidation of the cyclopropane substrates. The charge distribution of phenylcyclopropane radical cation has been studied by DFT calculation (Fig. 5b). The results show that distal C atom in cyclopropane motif possess partial positive charge and are the potential nucleophilic attack sites. Considering the reaction between benzyl radical and dioxygen was very fast (rate constant $2.8 \times 10^9$ L mol$^{-1}$ s$^{-1}$)[65], the reaction was conducted in the dioxygen atmosphere in order to trap the possible benzyl radical intermediate. The detection of the oxygenation products is highly inductive of the formation of the benzyl radical during the reaction (Fig. 5c). In addition, the existence of benzyl radical was also evidenced by the trapping experiment using BrCCl$_3$[66,67]. Further-more, the electrolysis of phenylcyclopropane and Et$_3$N·3HF in CH$_3$CN resulted in the isolation of amidation product **59**, which suggested the involvement of benzyl carbonium intermediate during the reaction (Fig. 5d)[68,69].

In conclusion, we have developed a electrochemical C–C bond cleavage of arylcyclopropanes, enabling 1,3-difunctionalization of arylcyclopropanes to yield 1,3-difluorination, 1,3-oxyfluorination, and 1,3-dioxygenation products. Neither additional oxidant or catalyst were needed in this transformation. Productive gram-scale 1,3-difluorination reaction was conducted by using stoichiometric amount of commercial available Et$_3$N·3HF as fluorine source. Mechanistic studies show that arylcyclopropane radical cation and benzyl carbonium play paramount role in this reaction. This study provides a simple strategy for constructing 1,3-difunctionalized molecules.

## Methods

**General procedure (2)**. An oven-dried undivided three-necked bottle equipped with a stir bar. The bottle was equipped with platinum plate (15 mm × 15 mm × 0.3 mm) as both the anode and cathode and then charged with argon gas in glove box. Phenylcyclopropane (0.5 mmol), Et$_3$N·3HF (1.2 mL) and PhCF$_3$ (4.8 mL) were added. The reaction mixture was stirred and electrolyzed at a constant current of 16 mA at 25 °C for 2 h. The reaction was diluted with water. The organic layer was extracted with CH$_2$Cl$_2$, dried with anhydrous Na$_2$SO$_4$, filtered, and concentrated under reduced pressure. The pure product was obtained by flash column chromatography on silica gel. Full experimental details can be found in the Supplementary Methods.

## Data availability

The authors declare that the data supporting the findings of this study are available within the article and its supplementary information files.

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

## Acknowledgements

This work was supported by the National Natural Science Foundation of China (22031008, 21520102003) and the Hubei Province Natural Science Foundation of China (2017CFA010). The Program of Introducing Talents of Discipline to Universities of China (111 Program) is also appreciated. The numerical calculations in this paper have been done on the supercomputing system in the Supercomputing Center of Wuhan University.

## Author contributions

A.L. and P.P. conceived the project and designed the experiments. P.P. developed the reactions, and contributed to the reaction scope and mechanistic studies. X.Y., K.Z., Z.L., and Y.C. contributed to the reaction scope and participated in the substrates synthesis. L.Z. and H.Z. discussed the results. H.Z. performed the theoretical calculation. P.P., H.Z., and A.L. wrote the paper, Supplementary methods, and related materials.

## Competing interests

The authors declare no competing interests.
