## [Peer Review File · Nature Communications]

REVIEWER COMMENTS

Reviewer #1 (Remarks to the Author):

This manuscript describes an electrochemical method for the C-C bond cleavage of arylcyclopropanes, leading to the formation of 1,3-difunctionalized arylpropanes. 1,3-Difluorinations, dialkoxylations and oxyfluorinations have been achieved in good yields by using different combinations of nucleophiles.

The anodic ring opening of arylcyclopropanes using methanol as nucleophile was described by Shono in the 1970s (J. Org. Chem. 1970, 35, 4157). The authors must know this early work, as it has been cited as reference 50 in the manuscript. However, the citation is misplaced and the work by Shono has not been properly acknowledged in the introduction.

The work presented in this manuscript represents a substantial advancement with respect to Shono's publication. The scope presented is very good, both for the arylcyclopropanes and nucleophiles, including useful fluorinations. Moreover, mechanistic insights are provided. The transformation can be useful and inspire further work in the area. Thus, this paper could be accepted for publication in Nature Communications after some modifications, listed below, are implemented:

- It is important that the authors properly describe and acknowledge the work by Shono in the introduction. This should include an appropriate description in the text and also in Figure 1, which should include the early anodic cyclopropane ring opening example and the reference.
- Some of the isolated products seem to contain high amounts of impurities according to the NMR spectra (e.g., compound 8 contains solvent). The authors should either provide cleaner spectra or, if some product is difficult to dry because it is volatile, the NMR purity should be mentioned.
- Some numbers in Figures 6a and 6b are difficult to see. The quality might be improved.

Reviewer #2 (Remarks to the Author):

In this manuscript, Lei and coworkers reported an electrochemical approach to functionalize phenylcyclopropanes for the synthesis of 1,3-difunctionalized molecules. Three different reactions, including 1,3-difluorination, 1,3-oxyfluorination and 1,3-dioxygenation, were reported.

Difunctionalization of phenylcyclopropane via ring opening reactions have been previously reported, but the advantage of using electrochemistry approach (this work) is to avoid the using of external oxidants. Mechanistic studies were conducted to probe the radical and carbocation intermediates involved in the reactions. Overall, the results reported in this manuscript are interesting and this work might be suitable for Nature Communications. However, the authors must address all of the following issues:

1. The substrate scope of the difluorination reaction seems rather limited. The authors mainly showed that electron-withdrawing groups could be tolerated under the reaction conditions, but the compatibility of electron-donating groups with this method has not been demonstrated. Could methoxy groups and other EDGs be tolerated on the phenyl rings in these reactions? In addition, only one heterocycle example (thiophene) was shown in Figure 2. Could other six-membered and five-membered heterocycles such as pyridines, pyrimidines, furans, pyrroles, etc. be tolerated under this difluorination conditions?

2. Another concern I have about these reactions is the synthetic utilities. The authors have shown that the difluorination reactions could be applied on the natural product scaffold. Yet, these examples are just simple attachment of a phenylcyclopropanes to natural products that do not have complex functional groups. The structures of these natural products might look large, but they not contain any complex functional groups.

In order to demonstrate the synthetic utilities of these reactions, the author should consider applying these methods to the synthesis of drug-like molecules. Fluorine-containing molecules are widely used as pharmaceuticals, and therefore the reported methods could potentially be a useful

way to synthesize the fluorinated derivatives of drug molecules.

3. It is unnecessary to separate the large-scale synthesis in a single table, which only contains four substrates. The authors should consider combining these data in Fig 2. – Fig. 4

This is also the case for Figure 4, which described the scope of di-methoxylation of phenyl-cyclopropanes with only 7 examples. The author should consider combining Fig 3 and Fig. 4 in one Figure.

4. It is interesting that cyclic ethers could also be used in the reactions for the construction of C-O bonds. The authors proposed a possible mechanism in the supporting information, but no evidences were provided to support the proposed mechanism. It is interesting that dimethoxyethane led to the formation of non-fluorinated product, while the other two ethers formed the fluorinated products. The authors should explain this result.

5. The yields of most products are moderate. What are the side products? Is the low yield due to the lower conversion of the starting materials or due to the formation of other side products?

6. The authors conducted cyclic voltammetry studies to explain the requirement of the phenyl groups on the cyclopropane rings for these reactions. Two onset potentials (without mentioning the reference electrodes in the main text) have been reported by the authors for two of the phenyl-cyclopropanes. However, it is very confusing how the authors were able to get these onsite potential data from the CV data.

7. The supporting information seems satisfactory and it is clear from the NMR spectra that these compounds could be isolated in high purity. In addition, the authors have provided detailed protocols, with pictures, for the set-up of the small-scale electrochemical reactions.

Could the authors also show the set-up, with pictures, of the large-scale synthesis? In addition, the resolution of the NMR spectra seems low and it is hard to see the splitting pattern of many HNMR spectra.

The splitting pattern for the ^{19}F NMR should also be reported. The ^{13}C NMR spectra should not be reported with two significant digits after the comma as it suggests an accuracy that is not met by the current NMR equipment.

8. High-resolution MS data have been reported for most of the compounds, but some of the compounds, for example, 18, 17, 15, 6, 19... do not have any MS data. Although these compounds have been previously reported, mass data should be reported for these molecules that were synthesized using a different method.

9. Grammar issues are throughout the entire manuscript; nearly every sentence of this manuscript has certain grammar mistakes. For example, the first sentence: "C-C bonds are the basic skeleton of organic compounds and widely existed in nature, it's deconstructive functionalization are meaningful" A comma should not be used to connect two complete sentences. The authors are strongly encouraged to have native speakers to polish this manuscript.

REVIEWER COMMENTS

Reviewer #1 (Remarks to the Author):

This manuscript describes an electrochemical method for the C-C bond cleavage of arylcyclopropanes, leading to the formation of 1,3-difunctionalized arylpropanes. 1,3-Difluorinations, dialkoxylations and oxyfluorinations have been achieved in good yields by using different combinations of nucleophiles.

Our response: Thanks for your comments.

The anodic ring opening of arylcyclopropanes using methanol as nucleophile was described by Shono in the 1970s (J. Org. Chem. 1970, 35, 4157). The authors must know this early work, as it has been cited as reference 50 in the manuscript. However, the citation is misplaced and the work by Shono has not been properly acknowledged in the introduction.

Our response: Thanks for your comments. Shono's work was introduced with citation in the revised second paragraph and Figure 1.

The work presented in this manuscript represents a substantial advancement with respect to Shono's publication. The scope presented is very good, both for the arylcyclopropanes and nucleophiles, including useful fluorinations. Moreover, mechanistic insights are provided. The transformation can be useful and inspire further work in the area. Thus, this paper could be accepted for publication in Nature Communications after some modifications, listed below, are implemented:

Our response: Thanks for your positive comments.

- It is important that the authors properly describe and acknowledge the work by Shono in the introduction. This should include an appropriate description in the text and also in Figure 1, which should include the early anodic cyclopropane ring opening example and the reference.

Our response: Thanks for your suggestion. The early anodic cyclopropane ring opening example was described in the revised second paragraph and in Figure 1.

- Some of the isolated products seem to contain high amounts of impurities according to the NMR spectra (e.g., compound 8 contains solvent). The authors should either provide cleaner spectra or, if some product is difficult to dry because it is volatile, the NMR purity should be mentioned.

Our response: Thanks for your suggestion. Some products (such as compound 2, 3, 5, 8, 21) will be removed under vacuum, the mass of these products were weighted and continuously decreased under vacuum. So we provided the NMR yield or GC yield instead of isolated yield in the Table 1 and Figure 2. For some compounds (especially for compound 8), it's hard to remove residual eluent without product loss.

- Some numbers in Figures 6a and 6b are difficult to see. The quality might be improved.

Our response: Thanks for your suggestion. We had provided high-resolution pictures in revised manuscript.

Reviewer #2 (Remarks to the Author):

In this manuscript, Lei and coworkers reported an electrochemical approach to functionalize phenylcyclopropanes for the synthesis of 1,3-difunctionalized molecules. Three different reactions, including 1,3-difluorination, 1,3-oxyfluorination and 1,3-dioxygenation, were reported. Difunctionalization of phenylcyclopropane via ring opening reactions have been previously reported, but the advantage of using electrochemistry approach (this work) is to avoid the using of external oxidants. Mechanistic studies were conducted to probe the radical and carbocation intermediates involved in the reactions. Overall, the results reported in this manuscript are interesting and this work might be suitable for Nature Communications. However, the authors must address all of the following issues:

Our response: Thanks for your comments.

1. The substrate scope of the difluorination reaction seems rather limited. The authors mainly showed that electron-withdrawing groups could be tolerated under the reaction conditions, but the compatibility of electron-donating groups with this method has not been demonstrated. Could methoxy groups and other EDGs be tolerated on the phenyl rings in these reactions? In addition, only one heterocycle example (thiophene) was shown in Figure 2. Could other six-membered and five-membered heterocycles such as pyridines, pyrimidines, furans, pyrroles, etc. be tolerated under this difluorination conditions?

Our response: The compatibility of electron-donating groups were further studied. Methoxy group (*para*-substituted or *meta*-substituted, 1,3,5-trisubstituted) and other EDGs (such as -SMe, -OPh, -NHBoc) on the phenyl rings could not be well tolerated in the difluorination reaction (Fig. 1), only trace amount product was detected by GC-MS for most cases. We had further study the conversion for *para*-methoxy group substituted phenylcyclopropane. Although the conversion was increased by prolong the reaction time (Fig. 2), we can't detect obvious product peak or by-products peaks through GC-MS.

Next, the compatibility of six-membered and five-membered heterocycles were also studied. Six-membered heterocycles such as pyridines, pyrimidines and quinolone were not amenable in this transformation, we can't detect desired product from GC-MS. The yield is low for furan-substituted cyclopropane because the selectivity is bad. Obvious difluorination isomer byproduct (the detailed structure can't be defined by MS) could be detected by GC-MS (Fig. 4).

Fig. 1

Fig. 2

Fig. 3

Fig. 4

2. Another concern I have about these reactions is the synthetic utilities. The authors have shown that the difluorination reactions could be applied on the natural product scaffold. Yet, these examples are just simple attachment of a phenylcyclopropanes to natural products that do not have complex functional groups. The structures of these natural products might look large, but they not contain any complex functional groups.

In order to demonstrate the synthetic utilities of these reactions, the author should consider applying these methods to the synthesis of drug-like molecules. Fluorine-containing molecules are widely used as pharmaceuticals, and therefore the reported methods could potentially be a useful way to synthesize the fluorinated derivatives of drug molecules.

Our response: Thanks for your suggestions. As exemplified by 1,3-oxyfluorination product **39** with *meta*-trifluoromethyl group substituted on the aryl ring, this 1,3-oxyfluorination product is generated in 53% yield under the standard condition and potential to further convert to fluorinated Cinacalcet following the reported methods (*Tetrahedron Lett.* **1981**, *22*, 4239-4240, *ChemSusChem* **2020**, *13*, 3583 –3588).

Fig. 5

3. It is unnecessary to separate the large-scale synthesis in a single table, which only contains four substrates. The authors should consider combining these data in Fig 2. – Fig. 4

This is also the case for Figure 4, which described the scope of di-methoxylation of phenyl-cyclopropanes with only 7 examples. The author should consider combining Fig 3 and Fig. 4 in one Figure.

Our response: Thanks for your suggestions. Fig 2. and Fig. 4 were combined together in revised manuscript. Fig 3 and Fig. 4 were combined in one Figure in revised manuscript.

4. It is interesting that cyclic ethers could also be used in the reactions for the construction of C-O bonds. The authors proposed a possible mechanism in the supporting information, but no evidences were provided to support the proposed mechanism. It is interesting that dimethoxyethane led to the formation

of non-fluorinated product, while the other two ethers formed the fluorinated products. The authors should explain this result.

Our response: Thanks for your suggestions. We provide a more intuitive mechanistic diagram in the revised supporting information. As shown in the Fig. 6, when dimethoxyethane was used as the reaction substrate, the structure of the oxonium intermediate is shown in the Fig. 6 (Int 1). Int 1 could be nucleophilic attacked by fluoride to produce MeF. But when tetrahydrofuran is used as substrate, the C-F bond will be formed into final fluorinated products.

Fig. 6

5. The yields of most products are moderate. What are the side products? Is the low yield due to the lower conversion of the starting materials or due to the formation of other side products?

Our response: Thanks for your suggestions. The conversion was high for most cases, and there were some unknown side products generated during the reaction. For example, the reaction mixture for *para*-bromine substituted phenylcyclopropane was studied by GC-MS (Fig. 7). Trifluorination and oxyfluorination by-products were detected. Meanwhile, two unknown by-products were also detected.

Fig. 7

6. The authors conducted cyclic voltammetry studies to explain the requirement of the phenyl groups on the cyclopropane rings for these reactions. Two onset potentials (without mentioning the reference electrodes in the main text) have been reported by the authors for two of the phenyl-cyclopropanes. However, it is very confusing how the authors were able to get these onsite potential data from the CV data.

Our response: Thanks for your comments. The reference electrode (Ag/AgCl/KCl_{sat} reference electrode) was mentioned in the revised manuscript. And half-peak potential instead of onset potential data were used in the revised manuscript.

7. The supporting information seems satisfactory and it is clear from the NMR spectra that these compounds could be isolated in high purity. In addition, the authors have provided detailed protocols, with pictures, for the set-up of the small-scale electrochemical reactions.

Could the authors also show the set-up, with pictures, of the large-scale synthesis? In addition, the resolution of the NMR spectra seems low and it is hard to see the splitting pattern of many HNMR spectra.

The splitting pattern for the ¹⁹F NMR should also be reported. The ¹³C NMR spectra should not be reported with two significant digits after the comma as it suggests an accuracy that is not met by the current NMR equipment.

Our response: Thanks for your comments. The set-up of the large-scale synthesis was provided in revised manuscript. HNMR spectra with higher resolution for all compounds were provided in revised supporting information. ¹⁹F NMR were re-done and the splitting pattern were reported for all compound in revised supporting information. All ¹³C spectra were reported with one significant digits after the comma in revised supporting information.

8. High-resolution MS data have been reported for most of the compounds, but some of the compounds, for example, 18, 17, 15, 6, 19.... do not have any MS data. Although these compounds have been previously reported, mass data should be reported for these molecules that were synthesized using a different method.

Our response: Thanks for your suggestion. Mass data (for 6, 15, 17, 18 and 19) had been replenished in revised supporting information.

9. Grammar issues are throughout the entire manuscript; nearly every sentence of this manuscript has certain grammar mistakes. For example, the first sentence: “C-C bonds are the basic skeleton of organic compounds and widely existed in nature, it’s deconstructive functionalization are meaningful” A comma should not be used to connect two complete sentences. The authors are strongly encouraged to have native speakers to polish this manuscript.

Our response: Thanks for your suggestion. We had carefully checked and revised the grammar issues of this manuscript.

REVIEWERS' COMMENTS

Reviewer #1 (Remarks to the Author):

The authors have correctly addressed the comments. The paper can be published

Reviewer #2 (Remarks to the Author):

With all the new experiments and detailed explanations, the authors have very nicely addressed all of my concerns. Therefore, I recommend the publication of this work on Nature Communications.